# MoVA: Adapting Mixture of Vision Experts to Multimodal Context

**Zhuofan Zong**[1,2,*]    **Bingqi Ma**[2,*]    **Dazhong Shen**[3]    **Guanglu Song**[2]

**Hao Shao**[1]    **Dongzhi Jiang**[1]    **Hongsheng Li**[1,3,4,†]    **Yu Liu**[2,†]

[1]CUHK MMLab    [2]SenseTime Research    [3]Shanghai AI Laboratory    [4]CPII under InnoHK

## Abstract

As the key component in multimodal large language models (MLLMs), the ability of the visual encoder greatly affects MLLM's understanding on diverse image content. Although some large-scale pretrained vision encoders such as vision encoders in CLIP and DINOv2 have brought promising performance, we found that there is still no single vision encoder that can dominate various image content understanding, e.g., the CLIP vision encoder leads to outstanding results on general image understanding but poor performance on document or chart content. To alleviate the bias of CLIP vision encoder, we first delve into the inherent behavior of different pre-trained vision encoders and then propose the MoVA, a powerful and novel MLLM, adaptively routing and fusing task-specific vision experts with a coarse-to-fine mechanism. In the coarse-grained stage, we design a context-aware expert routing strategy to dynamically select the most suitable vision experts according to the user instruction, input image, and expertise of vision experts. This benefits from the powerful model function understanding ability of the large language model (LLM). In the fine-grained stage, we elaborately conduct the mixture-of-vision-expert adapter (MoV-Adapter) to extract and fuse task-specific knowledge from various experts. This coarse-to-fine paradigm effectively leverages representations from experts based on multimodal context and model expertise, further enhancing the generalization ability. We conduct extensive experiments to evaluate the effectiveness of the proposed approach. Without any bells and whistles, MoVA can achieve significant performance gains over current state-of-the-art methods in a wide range of challenging multimodal benchmarks. Codes and models are available at `https://github.com/TempleX98/MoVA`.

## 1    Introduction

Significant achievements in multimodal large language models (MLLMs) [1, 2, 3, 4, 5, 6, 7] have been witnessed due to their remarkable proficiency in solving open-world tasks. MLLMs acquire visual perception capacity while inheriting sophisticated reasoning abilities and knowledge from large language models (LLMs) [8, 9, 10]. The core idea behind MLLMs is projecting the vision representation into an LLM through a projector, facilitating a general-purpose multimodal understanding.

General multimodal understanding requires comprehending complex image contexts across various tasks and scenarios. The CLIP [11] vision encoder, pre-trained on large-scale image-text pairs with a contrastive loss, is widely considered as a flexible and popular choice among the latest leading MLLMs. However, training data and optimization target of the vision encoder determine its

---

*Equal contribution.
†Corresponding authors.

38th Conference on Neural Information Processing Systems (NeurIPS 2024).

Table 1: **Comparison of CLIP *vs.* state-of-the-art task-specific vision encoders.** Our evaluation criteria encompass a variety of dimensions: comprehensive benchmarks [16], text-oriented Visual Question Answering (VQA) [17, 18], general VQA [19], object hallucination [20], Referring Expression Comprehension (REC) [21], Referring Expression Segmentation (RES) [21], and medical VQA benchmark SLAKE [22]. We use the same data for each model.

| Vision Encoder | Task | MMB | DocVQA | ChartQA | GQA | POPE | REC | RES | SLAKE |
|---|---|---|---|---|---|---|---|---|---|
| CLIP [11] | Image-text Contrastive | **64.9** | 35.6 | 35.3 | 62.5 | 85.7 | 81.5 | 43.3 | 63.7 |
| DINOv2 [15] | Visual Grounding | 57.5 | 14.7 | 15.9 | **63.9** | 86.7 | **86.1** | 47.5 | 59.4 |
| Co-DETR [23] | Object Detection | 48.4 | 14.2 | 14.8 | 58.6 | **88.0** | 82.1 | 48.6 | 55.3 |
| SAM [24] | Image Segmentation | 40.7 | 13.9 | 15.0 | 54.0 | 82.0 | 79.2 | **49.3** | 57.7 |
| Pix2Struct [25] | Text Recognition | 41.9 | **57.3** | 53.4 | 51.0 | 78.1 | 59.2 | 32.2 | 44.0 |
| Deplot [26] | Chart Understanding | 36.2 | 40.2 | **55.8** | 48.1 | 75.6 | 51.1 | 27.0 | 44.5 |
| Vary [12] | Document Chart Parsing | 28.1 | 47.8 | 41.8 | 42.6 | 69.1 | 21.6 | 16.0 | 40.9 |
| BiomedCLIP [27] | Biomedical Contrastive | 40.0 | 15.3 | 16.8 | 50.8 | 76.9 | 57.8 | 27.4 | **65.1** |
| Plain fusion | - | 63.4 | 46.5 | 48.9 | 63.0 | 86.4 | 85.7 | 45.3 | 64.7 |
| MoVA | - | **65.9** | **59.0** | **56.8** | **64.1** | **88.5** | **86.4** | **49.8** | **66.3** |

inconsistent performance across tasks and scenarios, which will bias the generalization of multimodal large language models. For instance, MLLMs with a single CLIP vision encoder usually perform poorly on fine-grained tasks such as grounding and optical character recognition (OCR) [12]. Several works have attempted to incorporate extra state-of-the-art vision encoder experts to cope with the challenge. For example, both SPHINX [13] and MoF [14] integrate vision self-supervised learning features of DINOv2 [15] with MLLMs to enhance their visual grounding capabilities. Vary [12] introduces a new vision encoder expert for improved fine-grained document and chart parsing ability. Intuitively, it is necessary to explore the utilization of more task-specific vision encoder experts in MLLMs to promote model generalization across various domains.

We aim to start the exploration through empirical analysis of readily available vision experts. In particular, we focus on the multimodal capabilities of seven distinct state-of-the-art vision encoders based on LLaVA-1.5-7B [28]. The results in Table 1 reveal that MLLMs with these task-specific vision encoders achieve optimal performance in their respective area. Concurrently, we note that the plain fusion (concatenation) of vision encoder experts adopted in previous works [13] would not bring consistent improvement compared with the single task-specific vision expert in its proficient task. The inherent bias of each expert introduces biased information and leads to performance degradation in the plain fusion paradigm. For example, DINOv2 serves as an expert in visual grounding but performs poorly at text-oriented tasks. Representation of DINOv2 would be regarded as biased information in text-related scenarios so incorporating DINOv2 for these tasks would inevitably cause performance decrease. Consequently, a flexible method of vision encoder ensemble that dynamically activates and weights context-relevant task-specific vision experts can fully unleash the capacity of these models while avoiding model bias.

In this paper, we propose MoVA, a powerful MLLM, adaptively routing and fusing task-specific vision experts with a coarse-to-fine mechanism. Inspired by the powerful tool-use capabilities of LLM [29], the coarse-grained context-aware expert routing aims to employ LLM to select vision experts with strong relevance to the user's image and instruction from the expert model pool. Thanks to the strong generalization ability of LLM, we also can perform model routing for vision experts in open scenarios. The fine-grained expert fusion facilitates better extraction and integration of expert representations based on multimodal context. Specifically, the expert knowledge extractor in the mixture-of-vision-expert adapter (MoV-Adapter) will extract diverse task-specific knowledge from various vision experts through mixture-of-expert (MoE) cross-attention layers. The dynamic gating network can allocate precise expert-wise soft weights for the integration of extracted task-specific knowledge. Under the coarse-to-fine paradigm, we provide a flexible and effective manner of leveraging representation from experts based on multimodal context and model expertise, further enhancing the model generalization ability. As presented in Table 1, MoVA can preserve the optimal performance of a single relevant vision encoder by ignoring non-relevant experts on the GQA, POPE, and REC task. Besides, MoVA can further boost performances via the fine-grained fusion of multiple relevant vision experts on other tasks.

We conduct comprehensive experiments on various benchmarks to evaluate the effectiveness of MoVA, including MLLM benchmarks, visual question answering (VQA), visual grounding, and biomedical understanding. Without any bells and whistles, MoVA can achieve significant performance gains over current state-of-the-art methods.

The **contributions** of this work are three-fold: **(i)** By analyzing the performance of individual vision encoders versus the plain fusion of multiple encoders across various tasks, we reveal that the inherent bias of each vision encoder can diminish its generalization ability across other irrelevant domains. **(ii)** We propose MoVA, a powerful MLLM composed of coarse-grained context-aware expert routing and fine-grained expert fusion with MoV-Adapter. Based on multimodal context and model expertise, MoVA fully leverages representation from multiple context-relevant vision experts flexibly while avoiding biased information of irrelevant experts. **(iii)** We demonstrate the effectiveness of each component in MoVA by elaborate ablation studies. MoVA can achieve significant performance gains over state-of-the-art methods in a wide range of challenging benchmarks.

## 2 Related Work

Multimodal architectures [1, 3, 6, 30, 31, 32, 33, 34], optimization paradigm [35, 36], applications [37, 38, 39, 40, 41], and benchmarks [42, 43, 44, 45, 46, 47, 48] have recently achieved remarkable progress and garnered unprecedented attention within the academic community. Multimodal large language models (MLLMs) usually leverage the alignment from visual features to the linguistic feature space to achieve superior vision-language understanding capabilities based on off-the-shelf LLMs and vision encoders. CLIP vision encoder [11], which is trained in contrastive learning from billions of diverse image-text pairs [49, 50], is widely used among these works. For example, LLaVA [3] adopts an MLP projector to align visual tokens from the frozen CLIP vision encoder to the embedding layer of LLM. However, The representation from CLIP exhibits strong discriminative abilities in classification and recognition but only has limited performance on downstream tasks like location and relation understanding [51]. To break through this bottleneck, some works [4, 52] turn to unlock the CLIP vision encoder and further fine-tune the parameter with training data for downstream tasks. For instance, Qwen-VL [6] collected massive training data for grounding and OCR to jointly optimize the CLIP vision encoder and LLM. Recent works propose to involve an extra frozen vision encoder to enhance the performance of MLLMs. SPHINX [13] is one of the pioneers, where grounding capabilities have been significantly improved with the assistance of the DINOv2 [15]. Vary [12] introduces an extra encoder training on large-scale charts and document data to improve the performance on related downstream tasks.

## 3 MoVA Methodology

### 3.1 Overview

MoVA comprises five key components: **(i)** a pre-trained large language model (LLM) that generates accurate responses given the image tokens and instructions; **(ii)** a base vision encoder; **(iii)** vision experts that generate task-specific vision latent features; **(iv)** mixture-of-vision-expert adapter (MoV-Adapter) that performs fine-grained expert fusion based on the multimodal context.

As illustrated in Figure 1, MoVA consists of two stages: coarse-grained context-ware expert routing and fine-grained expert fusion with MoV-Adapter. First, our coarse-grained context-ware expert routing leverages the tool-use capabilities of LLM, routing the most appropriate experts from $N$ expert candidates via LLM to help the model answer the user's question. In the second stage, we turn to enhance the visual representation with a novel MoV-Adapter module in a fine-grained manner. More specifically, we leverage the mixture-of-expert (MoE) cross-attention layers to extract the task-specific knowledge of representations from chosen experts. Meanwhile, the dynamic gating network in MoV-Adapter can allocate soft weights to the extracted knowledge of each expert according to the input image and instruction. Then the extracted knowledge can be effectively integrated into the foundational representation of the base vision encoder. Finally, the enhanced visual representation with instruction tokens is fed to the LLM to generate an accurate response. In Section 3.2 and Section 3.3, we will focus on our core contributions, the context-aware expert routing strategy, and the expert fusion with MoV-Adapter. In Section 3.4, we will introduce the training process.

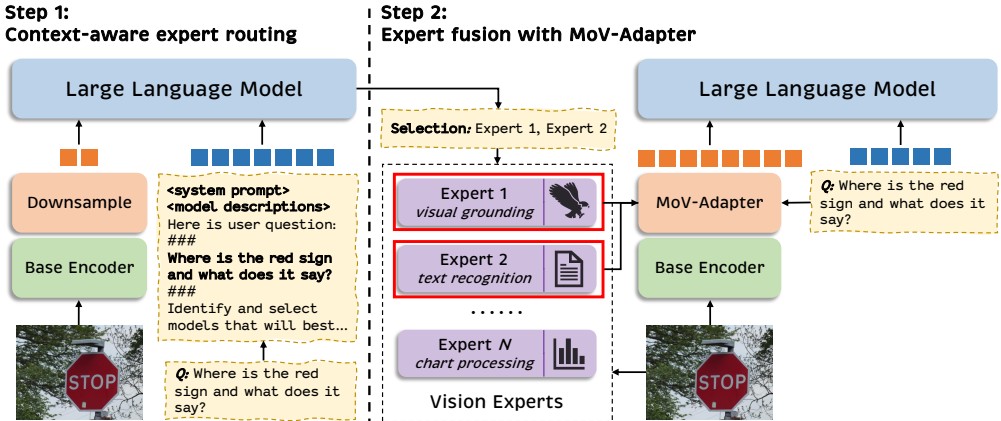

Figure 1: **The pipeline of MoVA.** MoVA performs coarse-to-fine routing to solve a given question. The coarse context-aware expert routing is performed in the first stage to select context-relevant experts. Next, we adopt the MoV-Adapter to extract and fuse the task-specific knowledge from these selected experts in a fine-grained manner.

**Pretrained Vision Encoders and LLM.** The vision encoders in MoVA consist of a base encoder and multiple task-specific vision encoder experts. We choose the pre-trained CLIP ViT-L-336px as the base encoder. Our vision experts include several state-of-the-art task-specific encoders: DINOv2, Co-DETR, SAM, Pix2Struct, Deplot, Vary, and BiomedCLIP. The corresponding expertise is presented in Table 1. For example, both Pix2Struct and Vary will be used when the user asks the MLLM to scan the document image. MoVA is flexible and easy to generalize to all decoder-only LLMs. We mainly consider Vicuna-7B [8], Llama3-8B [3], and Yi-34B [53] as our language models in this work.

## 3.2    Coarse-grained Context-aware Expert Routing

**Pipeline of Context-aware Routing.** The context-aware expert routing strategy aims to employ the impressive tool-use capacity of LLM to select vision experts with strong relevance to the user's image and instruction from a model pool. Specifically, we perform the context-aware expert routing in three steps during inference. First, the input image, user questions, and descriptions of expert models are converted into appropriate instructions that prompt the MLLM to perform expert selection. An example of the prompt instruction input and selection output is shown in Table 2. Such a routing task does not require image details and high-resolution input images, hence we directly downsample the base encoder's visual feature to obtain a coarse image embedding (*e.g.*, 144 image tokens). The downsampled image tokens and instruction tokens are then fed to the LLM as inputs. Finally, the LLM generates the output text and we parse it to determine which vision expert should be selected for fine-grained knowledge extraction in the second stage. For instance, as depicted in Table 2, the LLM directly outputs the option's letter of DINOv2 and Pix2Struct, thus we only utilize them for the subsequent extraction. During training, we do not perform context-aware expert routing and replace the routing outputs with our routing annotations to improve efficiency.

**Routing Data Construction.** Compared with other MLLMs, MoVA requires additional routing annotations. We first introduce the formal definition of the data structure for an unambiguous understanding of the routing data. The data structure for expert routing introduces additional routing annotation $\mathcal{R}$ to the conventional multimodal data $(\mathcal{I}, \mathcal{Q}, \mathcal{A})$. Here, $\mathcal{I}$ represents the image, $\mathcal{Q}$ and $\mathcal{A}$ refer to the question-answer pair, and $\mathcal{R}$ refers to the expert set which contains the most appropriate ones to solve this question. Then the construction process for routing data can be formulated as $(\mathcal{I}, \mathcal{Q}, \mathcal{A}) \rightarrow \mathcal{R}$, with the primary objective being to derive vision experts that optimally align with the sample $(\mathcal{I}, \mathcal{Q}, \mathcal{A})$. Intuitively, the language modeling loss can serve as an effective metric for evaluating how a data sample aligns with the vision expert. Specifically, we can reuse the LLaVA-1.5-7B models with various vision encoders presented in Section 1 to perform loss computation. Here, we denote the model with the base encoder as $\mathcal{M}_0$ and the model with $j$-th expert among $N$ experts as

---

[3]https://github.com/meta-llama/llama3

Table 2: One example of the instruction-following data for context-aware expert routing. We present the multimodal inputs in the top block and the language response in the bottom block. The detailed model descriptions are released in the Appendix.

---

**Routing Prompt Input**
You are a helpful assistant router. Based on the visual content, questions, and model pool the user provides, you need to consider the expertise of these models to select the most 3 suitable models to help you answer the questions. Answer with the model's letter from the given choices directly. If no models are selected, just answer 'none'.
Model pool:
A. <DINOv2 model description>
B. <Co-DETR model description>
C. <SAM model description>
D. <Pix2Struct model description>
E. <Deplot model description>
F. <Vary model description>
G. <BiomedCLIP model description>
Question:
Where is the red sign and what does it say?

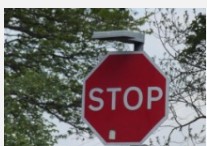

---

**Routing Prompt Output**
A, D

---

$\mathcal{M}_j$. For the $i$-th sample $(\mathcal{I}_i, \mathcal{Q}_i, \mathcal{A}_i)$, we send it to models $\{\mathcal{M}_j | j \in \{0, 1, \ldots, N\}\}$ and calculate the language modeling loss $\{\mathcal{L}_{i,j} | j \in \{0, 1, \ldots, N\}\}$. The $j$-th expert is regarded as a useful expert for the $i$-th sample only if $\mathcal{L}_{i,j} < \mathcal{L}_{i,0}$ and will be added to the routing set $\mathcal{R}_i$. Note that we only keep up to 3 vision experts to avoid computation costs brought by too many additional experts. All the routing annotations of our training data are generated offline. We can directly parse and input these offline results to the subsequent expert fusion component during training.

**Routing Data Augmentation.** To preserve the expert routing robustness and generalization ability in open scenarios, we only randomly select 2K samples for training, remove the model name in model description, and rewrite the model descriptions using ChatGPT [54] for each expert. We also shuffle the model pool and randomly truncate the model pool during training.

### 3.3 Fine-grained Expert Fusion with MoV-Adapter

We propose the MoV-Adapter to facilitate fine-grained expert representation extraction and integration based on multimodal context. As shown in Figure 2, the MoV-Adapter consists of $L$ adapter blocks and a text encoder. Each block contains an expert knowledge extractor, a dynamic gating network, and a transformer block. For the $i$-th block, the input feature is denoted as $\mathbf{X}^i \in \mathbb{R}^{C \times H \times W}$ and we take the CLIP base encoder feature $\mathbf{X} \in \mathbb{R}^{C \times H \times W}$ as the input feature $\mathbf{X}^1$ of the first block. We use $\mathbf{G}$ to indicate the indices of chosen $K$ experts. The expert feature set is $\{\mathcal{F}_j | j \in \mathbf{G}\}$. The final output feature of $L$ adapter blocks is $\mathbf{X}^{L+1}$. Additionally, we apply two residual blocks [55] with an average pooling to $\mathbf{X}^{L+1}$ to obtain a coarser image feature $\mathbf{X}^{L+1}_{out} \in \mathbb{R}^{C \times \frac{H}{2} \times \frac{W}{2}}$, which is further connected to the LLM text embedding space by an MLP layer.

**Text Encoder.** We introduce a pre-trained BERT as the text encoder to extract language context information from the user's instruction. We take the [CLS] token from the output of the text encoder as the text token $\mathbf{X}_T \in \mathbb{R}^{C_T}$. It is worth noting that all the adapter blocks share the same text token.

**Expert Knowledge Extractor.** We adopt $N$ cross-attention layers as the expert knowledge extractor to achieve efficient knowledge extraction. Note that only the expert features $\{\mathcal{F}_j | j \in \mathbf{G}\}$ and their corresponding cross-attention layers are involved in the extraction. For each selected expert feature $\mathcal{F}_j \in \mathbb{R}^{C_j \times H_j \times W_j}$, we first align its resolution to $\mathbf{X}^i$ with bilinear interpolation:

$$\hat{\mathcal{F}}_j = \text{Interpolate}(\mathcal{F}_j, H, W). \tag{1}$$

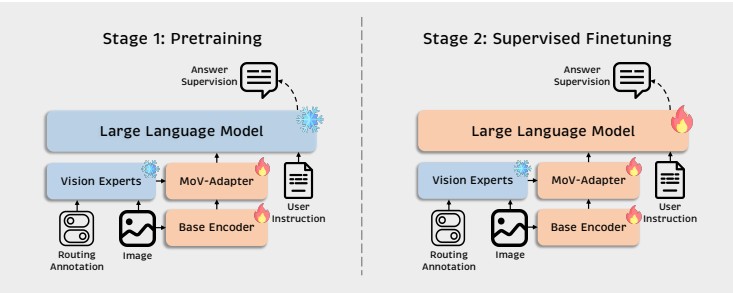

Figure 3: **The training strategy of MoVA.** We enhance the task-specific knowledge extraction capacity in the first stage. Then, we excite model multimodal capacities in the last stage.

For the $i$-th MoV-Adapter block and the $j$-th cross-attention layer, we take input feature $\mathbf{X}^i$ as query, and the aligned expert feature $\hat{\mathcal{F}}_j$ as the key and value:

$$\mathbf{Y}_j^i = \mathbf{X}^i + \text{Attention}(\mathbf{X}^i, \hat{\mathcal{F}}_j). \tag{2}$$

**Dynamic Gating Network.** We employ a dynamic gating network to contribute to a fine-grained knowledge integration process for the conditional representation $\{\mathbf{Y}_j^i | j \in \mathbf{G}\}$. It is implemented with the softmax over the logits of an MLP layer, processing multimodal representation to generate expert-wise soft weight $\mathbf{P}^i \in \mathbb{R}^K$ for the output of each cross-attention layer in the extractor. Specifically, the input to the gating network is the concatenated vector of a visual token $\mathbf{X}_V^i \in \mathbb{R}^C$ and the text token $\mathbf{X}_T \in \mathbb{R}^{C_T}$. We obtain $\mathbf{X}_V^i$ with a global average pooling operation to $\mathbf{X}^i$. Then we concatenate them to compute the gating weights and the expert-wise outputs by computing the weighted sum:

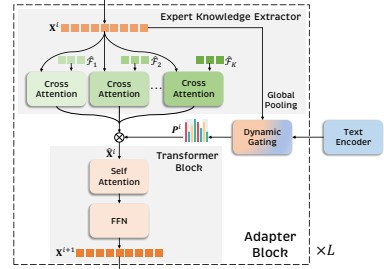

Figure 2: MoV-Adapter architecture.

$$\hat{\mathbf{X}}^i = \sum_{j \in \mathbf{G}} \mathbf{Y}_j^i \cdot \mathbf{P}_j^i, \tag{3}$$

where $\mathbf{P}_j^i \in (0, 1)$ is the soft weight for the $j$-th expert in the $i$-th block.

**Transformer Block.** The transformer block in the adapter block follows the vanilla design, consisting of a self-attention layer and an FFN layer. Taking the fused visual representation $\hat{\mathbf{X}}^i$, its output will serve as the input feature $\mathbf{X}^{i+1}$ for the next adapter block.

### 3.4 Training Paradigm

As shown in Figure 3, the training process of MoVA consists of pretraining and supervised finetuning.

**Pretraining.** To improve multimodal generalization, we first construct 15M visual instruction samples across diverse domains as the training data: **(i)** Image caption data that covers 4M randomly selected samples from DataComp-1B [56], ShareGPT4V-PT [52], and ALLaVA-4V [57]. **(ii)** Visual grounding and localization dataset that encompasses Objects365 [58], RefCOCO [21], VisualGenome [59], PointQA [60], and Flickr30K [61]. **(iii)** Chart understanding data that includes MMC-Instruction [62], Chart2Text [63], DVQA [64], and SciGraphQA [65]. **(iv)** Text recognition and document parsing data that covers LLaVAR-PT [66] and 3M English document images from Common Crawl [4]. **(v)** LLaVA-Med [67] for biomedical image understanding. During the pretraining phase, we only optimize the MoV-Adapter along with the base vision encoder while preserving the capabilities of the initial large language model. Meanwhile, we leverage the routing annotations generated via the method proposed in Section 3.2 to choose experts and ignore representations from irrelevant ones during training.

**Supervised Finetuning.** We utilize high-quality visual instruction tuning data that build upon LLaVA-665K [28] for finetuning. Additionally, we integrate several visual question answering

---

[4]https://commoncrawl.org

Table 3: **Performance comparison with current state-of-the-art frameworks on popular MLLM benchmarks.** PT and SFT indicate the number of multimodal training samples in pretraining and finetuning stage. #IMG means the number of image tokens processed by LLM.

| Model | LLM | PT | SFT | #IMG | MME | MMB | MMB$^{CN}$ | QBench | MathVista | MathVerse | POPE |
|---|---|---|---|---|---|---|---|---|---|---|---|
| *Proprietary MLLMs* | | | | | | | | | | | |
| Qwen-VL-Plus [6] | – | – | – | – | – | 66.2 | 68.0 | 66.0 | 43.3 | 11.8 | – |
| Qwen-VL-Max [6] | – | – | – | – | – | 77.6 | 75.1 | 73.6 | 51.0 | 24.8 | – |
| Gemini-Pro [79] | – | – | – | – | – | 73.6 | 74.3 | 68.2 | 45.2 | 22.3 | – |
| GPT-4V [54] | – | – | – | – | – | 75.8 | 73.9 | 74.5 | 49.9 | 38.3 | – |
| *Open-source MLLMs* | | | | | | | | | | | |
| Qwen-VL [6] | Qwen-7B | 1.4B | 50M | 256 | – | 38.2 | 7.4 | 59.4 | – | – | – |
| Qwen-VL-Chat [6] | Qwen-7B | 1.4B | 50M | 256 | 1488/361 | 60.6 | 56.7 | – | – | – | – |
| LLaVA-1.5 [28] | Vicuna-7B | 558K | 665K | 576 | 1511/316 | 64.3 | 58.3 | 58.7 | – | 14.3 | 85.9 |
| LLaVA-1.5 [28] | Vicuna-13B | 558K | 665K | 576 | 1531/295 | 67.7 | 63.6 | 62.1 | 27.6 | 17.0 | 85.9 |
| mPLUG-Owl2 [80] | LLaMA2-7B | 348M | 1.2M | 64 | 1450/– | 64.5 | – | 62.9 | – | 4.6 | 85.8 |
| SPHINX-2k [13] | Vicuna-13B | 115M | – | 2880 | 1471/327 | 65.9 | 57.9 | – | 27.8 | – | 87.2 |
| LLaVA-NeXT [81] | Vicuna-7B | 558K | 760K | 2880 | 1519/332 | 67.4 | 60.6 | – | 34.6 | – | 86.5 |
| LLaVA-NeXT [81] | Hermes-Yi-34B | 558K | 760K | 2880 | **1631**/397 | 79.3 | 79.0 | – | **46.5** | 23.4 | 87.7 |
| **MoVA** | Vicuna-7B | 15M | 1.6M | 576 | 1562/371 | 70.4 | 63.7 | 69.3 | 37.6 | 19.7 | 88.6 |
| **MoVA** | Llama3-8B | 15M | 1.6M | 576 | 1596/348 | 75.3 | 67.7 | **70.8** | 37.7 | 21.4 | **89.3** |
| **MoVA** | Hermes-Yi-34B | 15M | 1.6M | 576 | 1603/**455** | **81.3** | **79.0** | 70.7 | 44.3 | **23.7** | 88.8 |

datasets across various domains, such as DocVQA [17], ChartQA [18], InfographicVQA [68], AI2D [69], ST-VQA [70], TextVQA [71], SynthDoG-en [72], Geometry3K [73], PGPS9K [74], Geo170K [75], RefCOCO, LLaVA-Med, VQA-RAD [76], and SLAKE [22]. We also encompass equivalent comprehensive captions [52, 57, 77, 78] generated by the advanced GPT4-V [54] for improved world knowledge. Apart from the above instruction tuning data, we convert the selected 2K routing annotations to instructions and incorporate them into the training data. In the supervised fine-tuning stage, only task-specific vision experts are frozen and we jointly optimize other components. The objective of supervised fine-tuning is to align the visual representation and the embedding of LLM, boosting its visual instruction-following capabilities.

## 4 Experiments

### 4.1 Implementation Details

As mentioned in Section 3.4, our training pipeline consists of two stages. In the pretraining stage, we use the AdamW optimizer with an initial learning rate of $2\times10^{-4}$, a batch size of 1024, and train the model for 1 epoch. We jointly finetune the weights of all components except additional vision experts with a batch size of 128 and an initial learning rate of $2\times10^{-5}$ during supervised finetuning. We use 3 transformer blocks ($L=3$) in the MoV-Adapter and its hidden dimension is 1024, which is consistent with the base vision encoder CLIP. The input resolution of the base vision encoder is $672\times672$. Two residual blocks with an average pooling are employed in the MoV-Adapter to reduce the number of output image tokens from 2304 to 576. For the experiment performed in Table 1, we follow the default setting of LLaVA-1.5 but incorporate several additional datasets, including DocVQA [17], ChartQA [18], RefCOCO [21], LLaVA-Med [67], VQA-RAD [76], and SLAKE [22]. More details about vision experts, ablations, and analysis are available in Appendix A.1 and A.3.

### 4.2 MLLM Benchmarks

We empirically analyze the multimodal capacity and generalization ability of MoVA on a wide range of challenging MLLM benchmarks in Table 3. This comprehensive assessment is conducted on MME [82], MMBench [16], QBench [83], MathVista [84], MathVerse [85], and POPE [20]. Compared to other open-source MLLMs with similar model complexity, MoVA with Vicuna-7B achieves the best performance across 7 MLLM benchmarks while offering a more favorable balance between training efficiency and performance. For instance, MoVA-7B surpasses the recent state-of-the-art LLaVA-NeXT-7B [81] with a dynamic high resolution design, processing only 20% image tokens. Furthermore, we adopt Hermes-Yi-34B [86] as the LLM to validate the scaling property of MoVA. As depicted in Table 3, the performance of MoVA-34B is on par with popular proprietary

Table 4: **Performance comparison on VQA benchmarks**. We present the number of model parameters of each MLLM for a clear complexity comparison. * denotes zero-shot evaluation.

| Model | LLM | Params | General VQA | | | Text-oriented VQA | | | |
| | | | VQA$^{v2}$ | GQA | SQA$^{I}$ | TextVQA | ChartQA | DocVQA | AI2D |
|---|---|---|---|---|---|---|---|---|---|
| *Generalist models* | | | | | | | | | |
| Qwen-VL [6] | Qwen-7B | 10B | 79.5 | 59.3 | 67.1* | 63.8 | 65.7 | 65.1 | 62.3 |
| Qwen-VL-Chat [6] | Qwen-7B | 10B | 78.2 | 57.5 | 68.2* | 61.5 | 66.3 | 62.6 | 57.7 |
| LLaVA-1.5 [28] | Vicuna-7B | 7B | 78.5 | 62.0 | 66.8* | 58.2* | – | – | – |
| LLaVA-1.5 [28] | Vicuna-13B | 7B | 80.0 | 63.3 | 71.6* | 61.3* | – | – | – |
| SPHINX-2k [13] | Vicuna-13B | 16B | 80.7 | 63.1 | 70.6* | 61.2 | – | – | 65.1 |
| Vary-base [12] | Qwen-7B | 7B | – | – | – | – | 65.3 | 76.3 | – |
| CogAgent [87] | Vicuna-7B | 18B | 83.7 | – | – | 76.1 | 68.4 | 81.6 | – |
| *Specialist models* | | | | | | | | | |
| Pix2Struct-Large [25] | – | 1.3B | – | – | – | – | 58.6 | 76.6 | 42.1 |
| PALI-X-55B [88] | – | 55B | **86.0** | – | – | 71.4 | 70.9 | 80.0 | 81.2 |
| **MoVA** | Vicuna-7B | 10B | 83.5 | 64.8 | 74.4* | 76.4 | 68.3 | 81.3 | 74.9 |
| **MoVA** | Llama3-8B | 11B | 83.5 | 65.2 | 74.7* | 77.1 | 70.5 | 83.4 | 77.0 |
| **MoVA** | Hermes-Yi-34B | 38B | 82.3 | 63.9 | **79.0*** | **77.8** | **73.8** | **84.2** | **83.0** |

Table 5: **Performance comparison (Acc@0.5) on RefCOCO REC task.** Specialists are specifically designed for the grounding task or finetuned on RefCOCO data.

| Type | Model | RefCOCO | | | RefCOCO+ | | | RefCOCOg | |
| | | val | test-A | test-B | val | test-A | test-B | val | test |
|---|---|---|---|---|---|---|---|---|---|
| Generalist | Shikra-13B [5] | 87.83 | 91.11 | 81.81 | 82.89 | 87.79 | 74.41 | 82.64 | 83.16 |
| | Ferret-13B [91] | 89.48 | 92.41 | 84.36 | 82.81 | 88.14 | 75.17 | 85.83 | 86.34 |
| | Qwen-VL [6] | 89.36 | 92.26 | 85.34 | 83.12 | 88.25 | 77.21 | 85.58 | 85.48 |
| | SPHINX-2k [13] | 91.10 | 92.88 | 87.07 | 85.51 | 90.62 | 80.45 | 88.07 | 88.65 |
| | **MoVA-7B** | 92.55 | 94.50 | 88.81 | 87.70 | 92.05 | 82.94 | 89.28 | 89.70 |
| | **MoVA-8B** | 92.18 | **94.75** | 88.24 | 88.45 | 92.21 | 82.82 | 90.05 | 90.23 |
| | **MoVA-34B** | **93.38** | 94.66 | **90.58** | **89.64** | **92.53** | **84.03** | **91.09** | **90.78** |
| Specialist | G-DINO-L [92] | 90.56 | 93.19 | 88.24 | 82.75 | 88.95 | 75.92 | 86.13 | 87.02 |
| | UNINEXT-H [93] | 92.64 | 94.33 | 91.46 | 85.24 | 89.63 | 79.79 | 88.73 | 89.37 |

MLLMs (*e.g.*, Gemini-Pro [79]) and outperforms Qwen-VL-Plus [6] on 5 MLLM benchmarks. For example, MoVA establishes new records on MMBench and MMBench-CN, even surpassing the GPT-4V [54]. These results suggest that the ensemble of vision experts with adaptive expert routing can serve as an effective dimension for MLLM model scaling.

### 4.3 Visual Question Answering

The evaluation results on VQA benchmarks are outlined in Table 4. We divide these benchmarks into general VQA benchmarks [89, 19, 90] and text-oriented VQA benchmarks [71, 18, 17, 69]. Thanks to the dynamic and efficient task-specific knowledge extraction, MoVA achieves state-of-the-art performances across diverse VQA benchmarks. For general VQA benchmarks, MoVA-7B outperforms SPHINX-2k [4] equipped with Vicuna-13B on VQAv2 [89] and GQA by 4.2% and 1.9%, respectively. Besides, MoVA shows its proficiency in text recognition in various scenarios, encompassing scene text, chart, document, and diagram. For instance, MoVA-7B catches up to the current state-of-the-art generalist CogAgent [87] with 18 billion parameters on these text-oriented benchmarks with smaller model size. The MoVA model with 38B parameters even surpasses the well-established specialist model PALI-X-55B [88] by clear margins. These outstanding performances demonstrate MoVA's robust generalization capabilities across diverse domains.

### 4.4 Visual Grounding

We conduct experiments on Referring Expression Comprehension (REC) benchmarks [21] to evaluate the visual grounding ability of MoVA. The results are presented in Table5. The performance of

Table 6: Comparisons on the biomedical VQA datasets.

| Model | VQA-RAD | | SLAKE | |
|---|---|---|---|---|
| | Open | Close | Open | Close |
| LLaVA-Med | 28.6 | 56.3 | 70.6 | 54.6 |
| LLaVA-1.5 | 35.3 | 68.9 | 73.1 | 63.7 |
| **MoVA** | **38.3** | **68.9** | **78.2** | **68.8** |
| LLaVA-Med (ft) | 61.5 | 84.2 | 83.1 | 85.3 |

Table 7: Results of component-wise ablation studies.

| Design | GQA | ChartQA | DocVQA |
|---|---|---|---|
| **MoVA** | **64.8** | **68.3** | **81.3** |
| Random routing | 63.1 | 60.4 | 71.6 |
| w/o routing | 63.4 | 62.5 | 73.7 |
| w/o MoV-Adapter | 62.7 | 65.2 | 77.1 |

Table 8: Results of $K$ varying from 1 to 3.

| $K$ | GQA | ChartQA |
|---|---|---|
| **Dynamic** | **64.8** | **68.3** |
| 1 | 64.0 | 64.9 |
| 2 | 63.5 | 66.7 |
| 3 | 63.2 | 67.4 |

Table 9: Comparisons of expert routing criteria.

| Design | #Models | POPE | GQA | ChartQA |
|---|---|---|---|---|
| **Separate** | 4 | 88.6 | **64.8** | 68.3 |
| Combination | 14 | 88.9 | 64.6 | 68.7 |

Table 10: Open-world expert routing results.

| Design | #Samples | Accuracy |
|---|---|---|
| **MoVA** | 2K | **92.5%** |
| MoVA | 5K | 12.5% |
| w/o Augmentation | 2K | 0% |
| MLP classifier | 2K | 0% |

Table 11: Performance of various MoV-Adapter variants.

| Design | MME$^P$ | MMB | POPE | GQA |
|---|---|---|---|---|
| **MoVA** | 1562 | **70.4** | **88.6** | **64.8** |
| 2 blocks | 1526 | 70.1 | 87.9 | 63.9 |
| 4 blocks | **1578** | 69.4 | 88.3 | 64.5 |
| Uniform gating | 1521 | 69.1 | 87.5 | 64.1 |

MoVA-7B is on par with the state-of-the-art specialist models that are elaborately designed for grounding tasks. For example, MoVA-7B achieves a score of 90.22% on RefCOCO+ val, which is 2.46% higher than the score of UNINEXT-H [93]. Our largest model MoVA-34B further pushes the performance bound of visual grounding on these benchmarks. These impressive results demonstrate MoVA's remarkable visual grounding capacity.

### 4.5 Medical Visual Question Answering

This experiment is conducted on popular medical VQA benchmarks VQA-RAD and SLAKE. We directly leverage the medical VQA evaluation metric adopted by LLaVA-Med. Each sample of VQA-RAD and SLAKE is observed only once during the training process of MoVA and LLaVA-1.5. For a fair comparison, we compare MoVA with the LLaVA-Med variant that is finetuned with only 1 epoch on the benchmark. The performance of the LLaVA-Med specialist that is fully finetuned on downstream tasks is also reported. As presented in Table 6, MoVA-7B consistently yields higher scores than LLaVA-Med and LLaVA-1.5, exhibiting its medical visual chat ability.

### 4.6 Ablation Study

**Component-wise analysis.** As presented in Table 7, we perform an ablation to thoroughly delve into the effect of each component. First, we try to replace the context-aware routing with random routing. Without task-relevant vision experts, the performance drops by a large margin, especially on the text-oriented VQA benchmarks. Removing context-aware routing to leverage all vision experts also leads to similar results. It proves that both these modifications introduce biased information from irrelevant vision experts due to the removal of context-aware routing. Then, we ablate the effectiveness of the MoV-Adapter by replacing it with simple linear layers. The removal of fine-grained expert feature fusion downgrades performance across all datasets. These results delineate that each component in MoVA can consistently yield significant gains.

**Number of activated experts.** In the context-aware routing phase, the number of activated experts $K$ is dynamic. We compare such a data-dependent design with other variations of constant $K$ in this experiment. As presented in Table 8, the overall performance of dynamic $K$ consistently outperforms other models with constant $K$. This reveals this dynamic implementation can fully exploit the task-specific knowledge of relevant experts while avoiding the incorporation of biased information.

**Criteria for choosing better experts.** To reduce the costs, our method only adopts $\binom{N}{1}$ models with $N$ various encoders to identify the better vision expert *separately*. However, we do not explicitly consider the combination of the chosen vision experts. In this experiment, we compare our method with another strategy that considers vision encoders combination and enumerates $\sum_{i=1}^{3} \binom{N}{i}$ models for routing data construction. Specifically, we set $N = 4$ and employ a smaller model pool of DINOv2, Co-DETR, Pix2Struct, and Deplot to reduce training costs. As shown in Table 9, our method achieves comparable performance while requiring much less models for data construction.

**Expert routing in open scenarios.** We develop 105 human-verified testing samples that should be answered using novel experts for the expert routing task. These novel experts encompass 7 vision models [94, 72, 95, 92, 96, 55, 97] on various computer vision tasks and each expert corresponds to 15 evaluation samples. We manually check the correctness of the expert routing result. As presented in Table 10, a lightweight network, such as a MLP classifier fails to generalize to this open-world setting. Besides, increasing the routing training samples and removing data augmentations also lead to severe performance degradation. The results demonstrate our coarse-grained context-aware routing preserves the generalization ability for expert routing in open scenarios.

**Adapter Design.** In this section, we conduct ablation studies on the design of the MoV-Adapter. As presented in Table 11, we compared the impact of using 2, 3, and 4 adapter blocks on the model's performance. We observed that the baseline with 3 blocks can achieve better performance than other settings. Then, we substituted our multimodal gating for uniform gating to investigate its effectiveness. Each of the experts is assigned the same soft weight in the uniform gating. We find uniform gating brings consistent performance drops in the test benchmarks. It indicates that the lack of the dynamic soft-weight harms the overall performance since it fails to perform precise knowledge extraction.

**Inference Analysis.** As illustrated in Figure 1, MoVA consists of two stages: coarse-grained context-ware expert routing and fine-grained expert fusion with MoV-Adapter. This two-stage inference pipeline can be further broken down into five steps: **(i)** Data preprocessing. We first process the input image with image processors and convert the input text into a token sequence with the LLM tokenizer. **(ii)** Base encoder forward. We extract the base image feature using the base CLIP encoder. Note that we only run the base encoder once since its output feature can be preserved and reused in the fourth step. **(iii)** LLM routing generation. We compress the base image features into 144 image tokens. The LLM generates a concise routing answer based on the compressed image feature and routing instruction. Vision experts and MoV-Adapter forward. **(iv)** According to the multimodal context and routing results generated in the previous step, we fuse vision features of the base encoder and activated experts in a coarse-to-fine manner. **(v)** LLM response generation. The LLM generates the final response given the fused vision features and user instructions. To investigate the inference efficiency of each step, we randomly select 200 images from the COCO val2017 dataset and adopt the common image caption instruction: *Describe this image.* The temperature for generation is 0. The latency is measured using bfloat16 and flash-attention 2 on an A100 80G GPU. We present the average inference latency of each step and show the average sequence length of the routing output and final response. The inference latencies for each step are 0.19s, 0.05s, 0.14s, 0.07s, and 10.24s, respectively. The average length of the routing output is 3.24 tokens, while the average length of the final response is 405.06 tokens. Compared to the LLM response generation (Step 5), the LLM expert routing (Step 3) generates much fewer output tokens and its latency is negligible (0.14s *v.s.* 10.24s). Therefore, our method does not bring significant inference costs.

## 5 Conclusion

In this paper, we reveal that the inherent bias of each vision encoder can diminish its generalization ability across other irrelevant domains by analyzing the performance of individual vision encoders versus the plain fusion of multiple encoders across various tasks. To deal with the problem, we propose MoVA, a powerful MLLM composed of coarse-grained context-aware expert routing and fine-grained expert fusion with MoV-Adapter. Based on multimodal context and model expertise, MoVA fully leverages representation from multiple context-relevant vision encoder experts flexibly and effectively while avoiding biased information brought by irrelevant experts. MoVA can achieve significant performance gains over current state-of-the-art methods in a wide range of benchmarks.

**Limitations.** We acknowledge some limitations in our paper that require attention. One limitation is the hallucination, which refers to the generation of text that appears plausible or coherent but is factually incorrect and misleading. This issue potentially presents in all powerful MLLMs. Additionally, the performance may be affected by failure cases of the context-relevant vision experts, leading to potential degradation. We plan to explore solutions for these limitations in future works.

## Acknowledgments and Disclosure of Funding

The work was supported by the National Key R&D Program of China under Grant 2021ZD0201300.

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

# A    Appendix / supplemental material

Table 12: **Vision expert model configurations of vision experts in MoVA.** Methods with * use a convolution layer to compress the output feature.

| Model | Params | Resolution | Width | Depth | Output shape |
|---|---|---|---|---|---|
| DINOv2-giant [15] | 1.1B | 518×518 | 1536 | 40 | 1536×37×37 |
| Co-DETR-large* [23] | 304M | 1280×1280 | 1024 | 24 | 256×80×80 |
| SAM-huge* [24] | 632M | 1024×1024 | 1280 | 32 | 256×64×64 |
| Pix2Struct-large [25] | 513M | 720×720 | 1536 | 18 | 1536×45×45 |
| Deplot-base [26] | 92M | 720×720 | 768 | 12 | 768×45×45 |
| Vary-base* [15] | 86M | 1024×1024 | 768 | 12 | 512×32×32 |
| BiomedCLIP-base [15] | 86M | 224×224 | 768 | 12 | 768×16×16 |

Table 13: **Introduction of datasets used in the MoV-Adapter pretraining stage.** The <class> placeholder represents the object category in the object detection task. The <expr> placeholder represents the expression in the REC task. The <bbox> placeholder denotes the bounding box coordinates. The <point> placeholder denotes the coordinate of a point. We directly use the original question as the instruction for MMC-Instruction and ScigraphQA.

| Task | Dataset | Task template |
|---|---|---|
| Image Caption | Datacomp [56] | Please describe this image.
Provide a one-sentence caption for the provided image. |
| | ShareGPT4V-PT [52] | Can you elaborate on the elements of the picture provided?
Write a detailed description of the given image. |
| | ALLaVA-4V [57] | Can you elaborate on the elements of the picture provided?
Write a detailed description of the given image. |
| Grounding and Localization | Objects365 [58] | Detect all objects among <class> in the image.
Perform object detection given the image within <class>. |
| | RefCOCO [21] | Locate the region this sentence describes: <expr>. Please provide the bounding box coordinates.
Please generate a short and spotlighted mention of the <bbox> part seen in the photo. |
| | Visual Genome [59] | Locate the region this sentence describes: <expr>. Please provide the bounding box coordinates.
Please generate a short and spotlighted mention of the <bbox> part seen in the photo. |
| | PointQA [60] | How many of these objects <bbox> in picture?
How many of these objects <point> in picture? |
| | Flickr30K [61] | Take a look at the image and give me the location details for any mentioned items.
Unravel the aspects of the image and give the bounding box for the mentioned items. |
| Chart Understanding | MMC-Instruction [62] | - |
| | Chart2Text [63] | What significant details and conclusions can be drawn from this chart?
Can you extract the data points in this image? |
| | ScigraphQA [65] | - |
| Document Parsing | LLaVAR-PT [66] | Report on any text that can be clearly read in the image.
Identify any text visible in the image provided. |
| | English Documents | Extract every piece of text from this image.
I request you to apply optical character recognition to this image. |
| Biomedical Understanding | LLaVA-Med [67] | Write a terse but informative summary of the picture.
Share a comprehensive rundown of the presented image. |

## A.1    Vision Experts

**Model Configuration.** We present the detailed model configurations of our task-specific vision experts in 12. We adopt the official checkpoint weights that are publicly available.

**Model Description.** The model descriptions used in the routing prompt are released in Table 17.

## A.2    Training Data Details

The training process of MoVA consists of two stages. In the Appendix, we present the training datasets with corresponding task templates of the first stage in Table 13. For the training data of the second stage, we follow the prompt format of LLaVA-1.5 [28]. The MoVA models with Vicuna-7B and LLama3-8B are pretrained using 64 A100 80G GPUs for 2 days, and finetuned using 32 A100 80G GPUs for 1 day. The MoVA with 34B LLM is pretrained using 128 A100 80G GPUs for 5 days and finetuned using 64 A100 80G GPUs for 2 days.

Table 14: Performance of various routing component.

| Design | #Samples | MME | MMB |
|---|---|---|---|
| **LLM** | 2K | **1562/371** | **70.4** |
| BERT | **1.6M** | 1520/326 | 68.8 |
| MLP | **1.6M** | 1483/305 | 68.1 |

Table 15: Results of various vision encoder combination for routing.

| Design | MMB | GQA | DocVQA |
|---|---|---|---|
| **CLIP** | **70.4** | 64.8 | **81.3** |
| +DINOv2 | 70.1 | **65.1** | 80.5 |
| +DINOv2+Pix2Struct | 69.5 | 64.4 | 80.9 |

Table 16: Effects of routing image tokens.

| #IMG | MMB | ChartQA |
|---|---|---|
| 144 | 70.4 | 68.3 |
| 256 | 69.8 | 68.4 |
| **576** | **70.6** | **68.7** |

## A.3 More Experiments

**Image segmentation.** In this experiment, we aim to investigate if task-specific knowledge can improve MoVA on the segmentation task. Therefore, we introduce a simple design to extend MoVA to segmentation tasks. Unlike segmentation generalists [98] that adopt an additional pixel decoder with high-resolution images for high-quality mask generation, we just formulate the referring segmentation task as sequential polygon generation [99]. We finetune MoVA and the baseline with a SAM-Huge [24] backbone on the RefCOCO referring segmentation datasets. MoVA achieves 57.1% gIoU on the testA benchmark, which is 2.6% higher than the 54.5% of baseline. This result indicates that MoVA is capable of exploiting task-specific knowledge to solve segmentation tasks.

**Effect of LLM for expert routing.** In this experiment, we investigate the effect of the LLM in our coarse-grained expert routing. As presented in Table 14, expert routing with LLM achieves the best performance. When the LLM is substituted for a lightweight MLP classifier and a BERT encoder, we need to increase the number of routing training samples from 2K to 1.6M to preserve model performance. Besides, both MLP classifier and BERT encoder fail to perform expert routing in open scenarios as stated in Table 10. Therefore, the strong tool-use capacity and generalization ability of LLM is critical to our flexible and effective expert routing.

**Vision encoder for expert routing.** In the coarse-grained expert routing, we only adopt the image feature of base vision encoder CLIP. As presented in Table 15, the method with CLIP achieves slightly better performance than other methods since such a routing task does not require elaborate expert knowledge. Besides, incorporating other vision experts with plain fusion also brings biased information and increases cost. To achieve a better trade-off between efficiency and performance, we only use CLIP for coarse-grained expert routing.

**Number of image tokens in expert routing.** We ablate the number of image tokens used in the expert routing stage. As shown in Table 16, routing with 144 tokens can achieve comparable performance to methods with more tokens. Considering the additional cost brought by processing more image tokens, we only use 144 tokens for routing.

**Data construction.** In this experiment, we analyze the effectiveness of our routing data construction method. The test split is constructed by randomly selecting 500 samples from the routing data. We first prompt GPT4-V [54] to determine whether the expert routing result is reasonable or not. These evaluation results are then manually checked and we finally compute the routing accuracy using the human-verified evaluation results. Our loss-driven data construction approach achieves 94.6% accuracy (473 samples are regarded as "reasonable" by GPT4-V), demonstrating its effectiveness.

## A.4 Qualitative Results

We apply MoVA to a wide range of real-world understanding and reasoning tasks to investigate its multimodal comprehension capacity. As presented in Figure 4 and Figure 5, MoVA successfully solves these diverse and complex cases thanks to the context-relevant expert knowledge.

## A.5 Potential Societal Impacts

**Positive impacts.** The positive impact is to use MoVA to solve various open-world tasks in real-world scenarios. For example, MoVA can serve as a useful assistant to help people improve work efficiency and answer their questions.

**Negative impacts.** The potential negative social impact is to use MoVA model to generate misleading and false contents. This issue potentially presents in all multimodal large language models. We will try to improve its safety in future works.

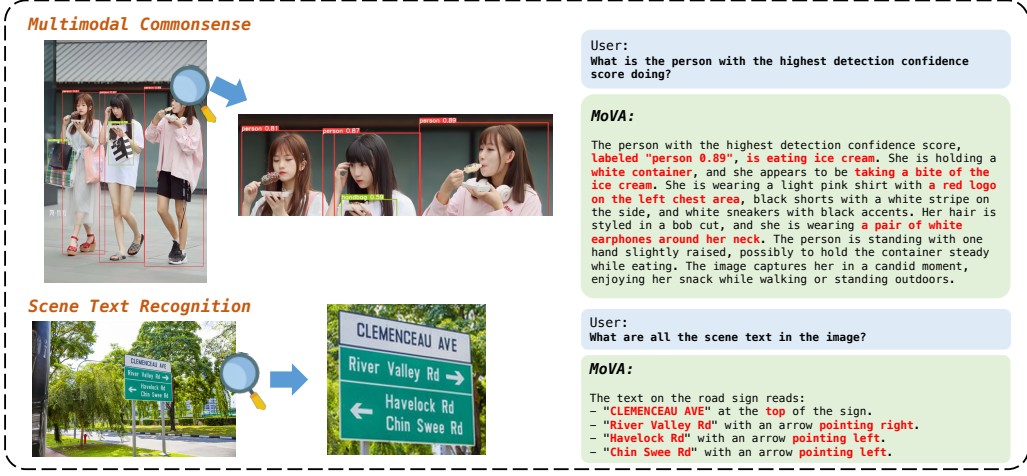

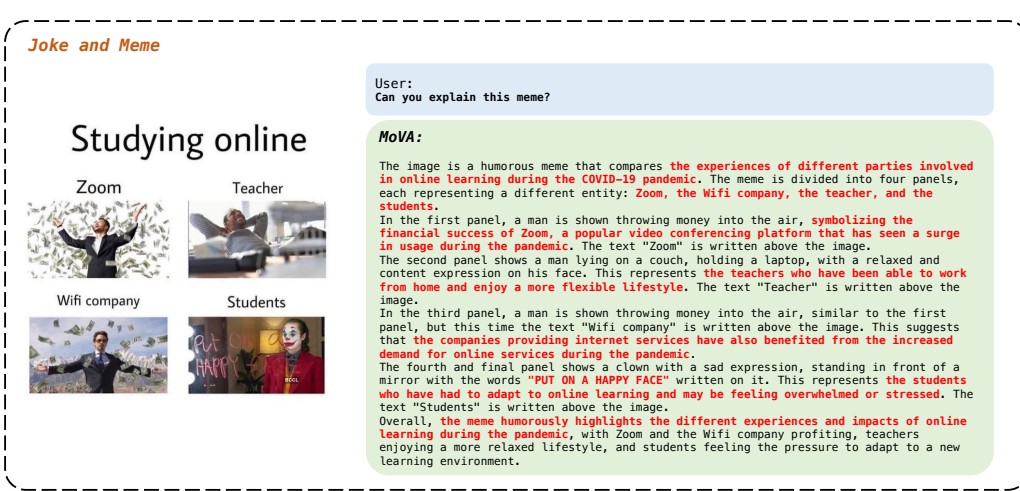

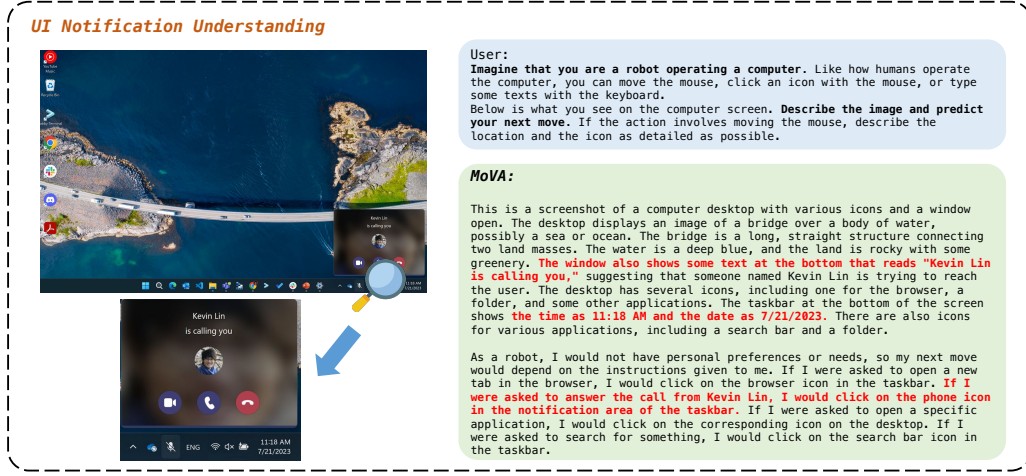

Figure 4: Qualitative multimodal understanding results of MoVA.

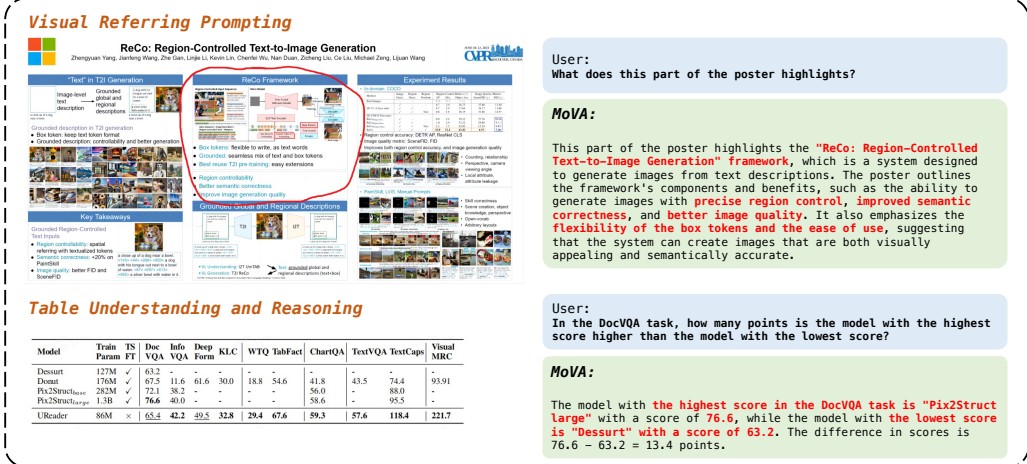

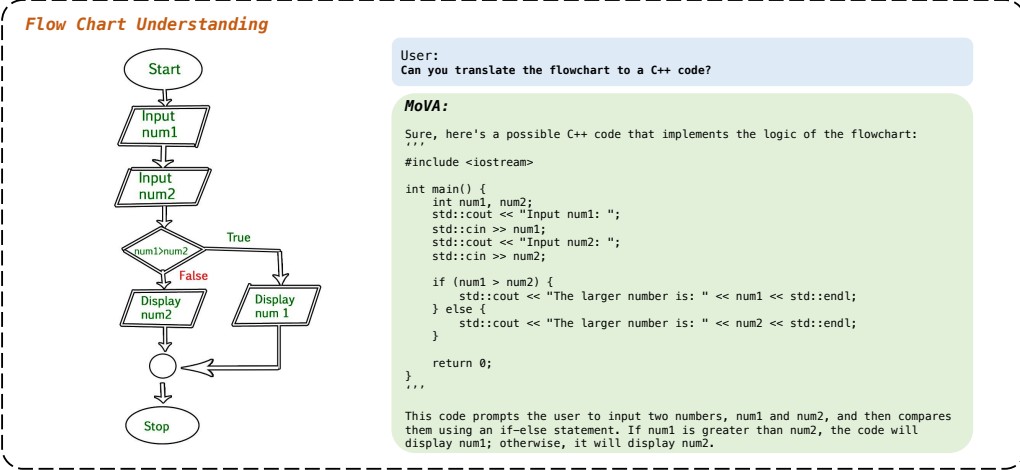

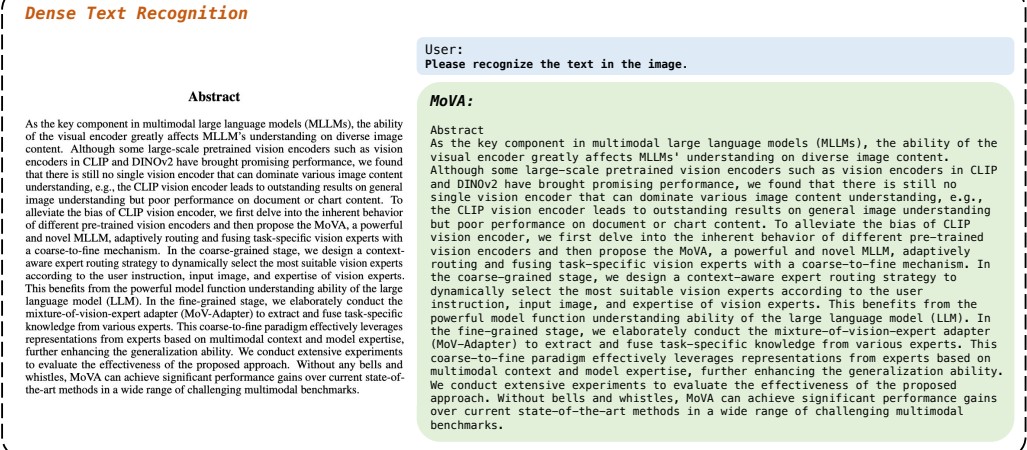

Figure 5: Qualitative multimodal understanding results of MoVA.

Table 17: **Model descriptions used in the context-aware expert routing.** We describe the pros and cons of each expert model in the routing prompt. We only present 3 captions for each expert here.

---

**DINOv2 description**

(1) This model demonstrates exceptional prediction capabilities across a range of image-related tasks, including image classification, object detection, segmentation, and image retrieval. The model leverages advanced self-supervised learning techniques to achieve high performance without relying heavily on labeled data.

(2) This model shows very strong prediction capabilities on tasks such as image classification, detection, segmentation, and image retrieval. However, it encounters challenges in accurately reading text within images.

(3) This model can effectively extract the accurate spatial and semantic information from natural images.

---

**Co-DETR description**

(1) This model is a state-of-the-art object detector pretrained on natural images. It can enable models to solve object-centric problems. Nonetheless, this model struggles with processing background elements in natural scenes.

(2) This model is a cutting-edge object detection model that can accurately detect objects in images. However, it struggles with identifying text in images.

(3) This model is a state-of-the-art object detector that can identify objects in images.

---

**SAM description**

(1) This model is an image segmentation model. This model can segment the precise location of either specific objects in an image or every object in an image.

(2) This model is a leading image segmentation framework and achieves strong zero-shot segmentation performance.

(3) This model is a promotable segmentation system with zero-shot generalization to unfamiliar objects and images.

---

**Pix2Struct description**

(1) This model excels in text recognition, achieving state-of-the-art text analysis results across distinct domains: documents, illustrations, user interfaces, natural images containing text, and images of charts.

(2) This model demonstrates exceptional proficiency in text recognition, delivering cutting-edge text analysis performance across various domains.

(3) This model can automate the extraction of information from scanned documents, making it easier to digitize and manage large volumes of paperwork.

---

**Deplot description**

(1) This model is a specialized model designed to achieve state-of-the-art plot and chart understanding performance.

(2) This model is a fine-tuned version of an existing text recognition model. It has been specifically trained to achieve superior performance in plot and chart understanding tasks.

(3) This model can help detect the text within the input document, diagram, and chart images.

---

**Vary description**

(1) This model can achieve more fine-grained vision perception for images with text, such as document-level Chinese/English OCR, book image to markdown or LATEX, Chinese/English chart understanding.

(2) This model can handle images with text effectively and accurately, enabling advanced tasks such as document OCR and chart understanding.

(3) This model can accurately process images with text, enabling tasks such as OCR. However, it cannot process natural images without text.

---

**BiomedCLIP description**

(1) This model is a foundation model designed for biomedical vision-language processing.

(2) This model is capable of biomedical images, such as chest X-ray and radiology images.

(3) This model is a state-of-the-art biomedical vision-language model. It has been shown to achieve significant improvements in biomedical image-text tasks.

