# OpenReview forum: "MoVA: Adapting Mixture of Vision Experts to Multimodal Context"
_NeurIPS.cc/2024/Conference — NeurIPS 2024 poster_

### Official Review · Reviewer_FVeU · 2024-07-09

**Soundness:** 3
**Presentation:** 4
**Contribution:** 2
**Rating:** 7
**Confidence:** 4

**Summary:**

The paper proposes a MoVA to extract and fuse task-specific knowledge from different visual encoder. Besides, the paper designs a coarse-to-fine mechanism, which dynamically selects the most suitable vision experts, then extracts and fuses task-specific knowledge from various experts. The experimental results are encouraging and validate the performance of the proposed method.

**Strengths:**

1. This paper proposes a new perspective, which is to improve the multi-task reasoning ability of MLLMs based on the different perception ability of the different encoders.
2. The paper is fluent and easy to understand.
3. The extensive experiments evaluate the effectiveness of the proposed approach.

**Weaknesses:**

1. The number of components in line 102 does not match the serial numbers.
2. During the inference phase, the process involves two rounds of LLM inference and two Base Encoder, which results in significant inference time and GPU memory usage. This poses a challenge for the practical application of MLLMs.
3. This is a well-completed article, but I am concerned about its novelty. The method seems similar to the adapter+agent approach described by Wang et al. [1]. Could you explain the differences and advantages of your method compared to theirs?

[1] Wang X, Zhuang B, Wu Q. Modaverse: Efficiently transforming modalities with LLMs. Proceedings of the IEEE/CVF Conference on Computer Vision and Pattern Recognition. 2024: 26606-26616.

**Questions:**

Why is the last row of MoVA results in Table 1 different from those in Table 3? In other words, The caption of Table 1 shows that the same data was used to train each model. Does this mean that all models used LLaVA-1.5 data? If so, I think this table will inspire the community and I would like to ask if you can release the checkpoints of these models.

**Limitations:**

1. As mentioned in the conclusion, the performance may be affected by failure cases of the context-relevant vision experts, leading to potential degradation. Additionally, it is influenced by the choice of encoder results in the first stage of inference.
2. The model's complex inference structure increases the inference burden.

---

> ### Author Rebuttal · Authors · 2024-08-07
>
> Dear Reviewer FVeU,
>
> Thanks for appreciating our work and your advice. We will address your concerns below.
>
> **Q1: The number of components in line 102 does not match the serial numbers.**
>
> There are four components in MoVA. We will fix this typo in the revised version.
>
> **Q2: The process involves two rounds of LLM inference and two Base Encoders, which results in significant inference time and GPU memory usage.**
>
> 1. As described in the general response, MoVA only runs the base encoder once during inference since its base vision feature can be reused.
> 2. Please refer to the Q3 of our global response.
>
> **Therefore, we think the inference cost is not the bottleneck for MoVA.**
>
> **Q3: The method seems similar to the adapter+agent approach described by Wang et al [1].**
>
> Both MoVA and ModaVerse are MLLM frameworks but they are different:
>
> | Model | Motivation | Setting | Method (input side) | Method (output side) |
> | - | - | - | - | - |
> | MoVA | We reveal that the feature fusion of fixed mixture of vision experts fails to achieve the optimal performance and leads to performance degradation if we do not consider the relations between model expertise and the current task. | MoVA focuses on the visual component design of large vision-language models | MoVA first uses the LLM to dynamically select $K$ experts that are most relevant to current task according to the model expertise and context. Then the vision features of selected experts and the base encoder are fused by an elaborate MoV-Adapter. | MoVA generates text outputs by a LLM. |
> | ModaVerse | ModaVerse operates directly at the level of natural language. It aligns the LLM’s output with the input of generative models, avoiding the complexities associated with latent feature alignments, and simplifying the multiple training stages of existing MLLMs into a single, efficient process. | ModaVerse is capable of interpreting and generating data in various modalities. | ModaVerse employs a single encoder to encode diverse multi-modal inputs and aligns them with LLM through a set of adapters. | ModaVerse treats the LLM as an agent and this agent produces a meta-response to activate generative models for generating the final response.
>
> **Advantages:**
>
> 1. ModaVerse uses a single encoder to encode inputs of various modalities. However, MoVA can dynamically select the most appropriate vision experts that are relevant to current inputs and tasks.
> 2. ModaVerse adopts a set of linear projection layers to align multimodal inputs and the LLM. In our paper, we propose the MoV-Adapter as the projector and demonstrate it can attain better performance than conventional linear layers.
>
> [1] ModaVerse: Efficiently Transforming Modalities with LLMs. CVPR 2024.
>
> **Q4: Why is the last row of MoVA results in Table 1 different from those in Table 3? Do all models in Table 1 use LLaVA-1.5 data? If so, I would like to ask if you can release the checkpoints of these models.**
>
> 1. The MoVA model in Table 3 uses more training data than the models in Table 1. The detailed data settings of Table 3 MoVA model are presented in the line 210.
> 2. As described in the line 229, we additionally incorporate several datasets (DocVQA, ChartQA, RefCOCO, LLaVA-Med, VQA-RAD, and SLAKE) to the LLaVA-1.5 665k SFT data to train the models in Table 1.
> 3. We will definitely release our codes and models to facilitate the community if the paper will be accepted.

---

> > ### Comment · Reviewer_FVeU · 2024-08-13
> >
> > Thank you for the response. It has addressed all of my concerns and I will raise my rating to 7.

---

> > > ### Author Response · Authors · 2024-08-13
> > > **Thanks for your comments**
> > >
> > > We sincerely thank the reviewer for the kind support of our work! We will incorporate the details into our final version.

---

### Official Review · Reviewer_gqRq · 2024-07-11

**Soundness:** 2
**Presentation:** 3
**Contribution:** 2
**Rating:** 5
**Confidence:** 5

**Summary:**

This paper proposes MoVA, a powerful and novel MLLM, adaptively routing and fusing task-specific vision experts with a coarse-to-fine mechanism. MoVA first leverages the tool-use capabilities of LLM, routing the most appropriate experts from expert candidates and then uses the MoV-Adapter module to enhance the visual representation.

**Strengths:**

1. The writing is easy to follow.
2. The idea of using LLM to route the most appropriate experts is interesting.

**Weaknesses:**

1. Is there an adapter structure in MoVA? Why do we call it MoV-Adapter?
2. The experiments use SFT data from different MLLMs should be aligned (the same). The comparison is not fair with different SFT data.
3. More vision encoder enhancement methods should be compared using the same SFT data, including

[1] VCoder: Versatile Vision Encoders for Multimodal Large Language Models.

[2] MoF (Eyes Wide Shut? Exploring the Visual Shortcomings of Multimodal LLMs)

[3] MouSi: Poly-Visual-Expert Vision-Language Models

[4] BRAVE: Broadening the visual encoding of vision-language models

4. The inference speed of LLM is very slow. I think MoVA using LLM as a router selection will decrease the speed of the whole process. How to solve this problem?

**Questions:**

In line 102, I just found four components, but the paper claims five.

**Limitations:**

The authors have adequately addressed the limitations.

---

> ### Author Rebuttal · Authors · 2024-08-07
>
> Dear reviewer gqRq,
>
> Thanks for your comments. We will address your concerns below.
>
> **Q1: Is there an adapter structure in MoVA? Why do we call it MoV-Adapter?**
>
> Similar to conventional MLLMs, MoVA comprises a pretrained LLM, a base vision encoder, multiple vision experts, and a projector that connects the LLM and vision encoders.
> We call this projector "MoV-Adapter" since it also can adapt output features of various activated vision experts to the base vision encoder feature space by elaborate expert knowledge extractor.
>
> **Q2: The comparison is not fair with different SFT data.**
>
> Please refer to the Q2 of our global response.
>
> **Q3: More vision encoder enhancement methods should be compared using the same SFT data.**
>
> We compare our method with VCoder [1], MoF [2], and MouSi [3] under the same SFT data and LLM (Vicuna-v1.5-7B) settings. We directly follow their corresponding evaluation benchmarks.
>
> | Model | Seg CS($\uparrow$) | Seg HS($\downarrow$) | Ins CS($\uparrow$) | Ins HS($\downarrow$) | Pan CS($\uparrow$) | Pan HS($\downarrow$) |
> | - | - | - | - | - | - | - |
> | VCoder-7B | 88.6 | 10.4 | 71.1 | 26.9 | 86.0 | 12.8 |
> | MoVA-7B | 89.1 | 10.0 | 73.8 | 24.7 | 87.6 | 11.4 |
>
> | Model | MMVP | POPE |
> | - | - | - |
> | MoF-7B | 28.0 | 86.3 |
> | MoVA-7B | 30.0 | 87.8 |
>
> | Model | VQAv2 | GQA | SQA | TextVQA | POPE | MMB | MMB_CN |
> | - | - | - | - | - | - | - | - |
> | MouSi-7B | 78.5 | 63.3 | 70.2 | 58.0 | 87.3 | 66.8 | 58.9 |
> | MoVA-7B | 80.1 | 65.1 | 71.2 | 68.5 | 87.8 | 67.2 | 58.4 |
>
> **Note that the authors of BRAVE [4] did not publicly release the codes, models, and data.** Due to the limited time of the rebuttal period, it is hard to reproduce its performance during rebuttal. We will add this comparison in the revised manuscript.
>
> [1] VCoder: Versatile Vision Encoders for Multimodal Large Language Models.
>
> [2] MoF (Eyes Wide Shut? Exploring the Visual Shortcomings of Multimodal LLMs)
>
> [3] MouSi: Poly-Visual-Expert Vision-Language Models
>
> [4] BRAVE: Broadening the visual encoding of vision-language models
>
> **Q4: Using LLM as a router will decrease the inference speed. How to solve this problem?**
>
> As highlighted by the general response, the expert routing only brings marginal inference costs thanks to our elaborate designs:
>
> 1. We use 144 compressed image tokens for expert routing. Note that the current state-of-the-art baseline LLaVA-NeXT employs the LLM to process 2880 image tokens.
> 2. Only the vision feature of the base vision encoder is used for expert routing.
> 3. During expert routing, we prompt the LLM to directly generate the option's letter of selected vision experts from the given model pool (e.g., "A" for DINOv2). As presented in the general response, the routing answer is short and **the average length of the routing output token sequence is only 3.24**.
>
> **Q5: In line 102, I just found four components, but the paper claims five.**
>
> This is a typo and it should be "comprises four key components". We will fix this typo in the revised version.

---

> ### Comment · Reviewer_gqRq · 2024-08-11
> **Question about the experiment results and novelty**
>
> Thanks for your response. I found the results you provided in the rebuttal period, including VCoder-7B, MoF-7B, MouSi-7B were the same as the results in their paper. **I strongly doubt that the authors did not use the same SFT data for a fair fine-tuning**. In our paper lines 230-232, the authors mention that they use additional SFT data. Please explain this, thanks!
>
>
> For Q4, I think comparing with LLava-Next is not appropriate, it is better to show the inference speed of MoE structure methods in [1-5].
>
> [1] VCoder: Versatile Vision Encoders for Multimodal Large Language Models.
>
> [2] MoF (Eyes Wide Shut? Exploring the Visual Shortcomings of Multimodal LLMs)
>
> [3] MouSi: Poly-Visual-Expert Vision-Language Models
>
> [4] BRAVE: Broadening the visual encoding of vision-language models
>
> [5] MoE-LLaVA: Mixture of Experts for Large Vision-Language Models
>
>
> Furthermore, after reading the reviews from Lntx, FVeU. I have the same question about the paper's novelty.
> About the general response Q1(1,2), **the typical MoE methods have the expert selection mechanism**, i.e. choosing K from N experts (e.g. Switch Transformer). For Q1(3), **it has been studied in Mousi [3] and is not the main contribution of this paper**.

---

> ### Author Response · Authors · 2024-08-11
> **Response to follow-up questions**
>
> Thanks for your follow-up questions. I'm delighted to address your concerns.
>
> **Q1: SFT data**
>
> We use the same SFT data as the original VCoder [1], MoF [2], and MouSi [3] implementations. Therefore, we directly report their performance scores in the paper.
>
> 1. VCoder. We employ the same COST dataset as the VCoder-7B to train the MoVA-7B model.
>
> 2. MoF and MouSi. Both of them use LLaVA-665k as the SFT data. To be specific, we institute the SFT data described in the line 210 for LLaVA-665k in our experiments.
>
> 3. The SFT data mentioned in the line 230 is only used for the model training in Table 1.
>
> [1] VCoder: Versatile Vision Encoders for Multimodal Large Language Models.
>
> [2] MoF (Eyes Wide Shut? Exploring the Visual Shortcomings of Multimodal LLMs)
>
> [3] MouSi: Poly-Visual-Expert Vision-Language Models
>
> **Q2: Inference comparison**
>
> We perform inference comparisons among MoVA, VCoder, MoF, and MoE-LLaVA [5] since the codes of both MouSi and BRAVE [4] are still publicly unavailable.
>
> For a fair comparison, we adopt the same number of output tokens (400 tokens). The benchmark setting is identical to the aforementioned benchmarking in the rebuttal.
>
> | Model | Average inference latency |
> | - | - |
> | VCoder-Vicuna-7B | 10.25s |
> | MoF-Vicuna-7B | 10.33s |
> | MoVA-Vicuna-7B | 10.47s |
> | MoE-LLaVA-Phi2-2.7B | 22.40s |
>
> The results reveal that the latency values of VCoder, MoF, and MoVA exhibit negligible differences. Moreover, MoE-LLaVA runs much slower than other frameworks even though it is equipped with a smaller LLM Phi2-2.7B.
>
> [4] BRAVE: Broadening the visual encoding of vision-language models
>
> [5] MoE-LLaVA: Mixture of Experts for Large Vision-Language Models
>
> **Q3: The typical MoE methods have the expert selection mechanism**
>
> Selecting $K$ experts from $N$ experts is a common practice of the mixture of experts (MoE) and it is not our contribution. Our paper focuses on **how to select $K$ vision experts** based on the current task in multimodal scenarios. Specifically, we explore how to leverage the reasoning ability of LLM and the prior knowledge of vision encoders to flexibly select the most $K$ appropriate vision experts. This idea has not been explored in the current mulitmodal community.
>
> **Q4: Comparison with MouSi**
>
> We compare our method with MouSi in the table:
>
> | Model | Motivation | Paradigm | Routing | Projector |
> | - | - | - | - | - |
> | MouSi | This paper proposes to leverage multiple experts to synergizes the capabilities of individual visual encoders. Experiments are conducted to demonstrate simple ensemble of multiple encoders can beat a single vision encoder. | Plain fusion | Experts are all activated. Relations between model expertise and the current task are ignored. | Linear layers |
> | MoVA | As discussed in the Q1 of global response, leveraging multiple encoders is not our contribution and our method is motivated by the limitations of these plain fusion methods. We observe the performance degradation occurs if we directly use and fuse all experts in a plain fusion manner. Such observation is not studied in MouSi. | Multimodal context-aware | We leverage the reasoning and tool-use ability of LLM to select the most appropriate experts considering model expertise, image, and instruction. | MoV-Adapter |
>
> **We will discuss and cite these works [1-4] in our revised paper.**
>
> [1] VCoder: Versatile Vision Encoders for Multimodal Large Language Models.
>
> [2] MoF (Eyes Wide Shut? Exploring the Visual Shortcomings of Multimodal LLMs)
>
> [3] MouSi: Poly-Visual-Expert Vision-Language Models
>
> [4] BRAVE: Broadening the visual encoding of vision-language models

---

> ### Comment · Reviewer_gqRq · 2024-08-13
> **About the inference speed**
>
> Thanks for your response. I wonder why expert selection (0.14s) in MoVA is so fast. I think the inference time for LLM should be longer.

---

> > ### Author Response · Authors · 2024-08-14
> > **Looking forward to your response**
> >
> > Dear Reviewer gqRq:
> >
> > We sincerely appreciate your time and efforts in reviewing our paper. We have provided corresponding responses and results, which we believe have covered your follow-up concerns. We hope to further discuss with you whether or not your concerns have been addressed. Please let us know if you still have any unclear parts of our work. If your concerns have been well addressed, please consider raising your rating, thanks.
> >
> > Best,
> >
> > Authors

---

> ### Author Response · Authors · 2024-08-13
> **Response to follow-up questions**
>
> Thanks for your question. We will address your concern below.
>
> We show the inference latency of each step in Q3 of our global response.
>
> | Step 1 | Step 2 | Step 3 | Step 4 | Step 5 | Routing output | Final response |
> | - | - | - | - | - | - | - |
> | 0.19s | 0.05s | 0.14s | 0.07s | 10.24s | 3.24 tokens | 405.06 tokens |
>
> The final response generation of LLM (Step 5) takes **$\frac{10.24}{405.06}=0.0253$ seconds to generate a token**, and the output speed of the LLM routing (Step 3) is  **$\frac{0.14}{3.24}=0.0432$ s/token**. Considering we adopt different prompt templates and image features for routing and final response generation, such speed value difference is reasonable. Specifically, we use the prompt template in Table 2 of our paper for expert routing and keep the same Vicuna template for response generation.
>
> Therefore, the expert routing is so fast since MoVA only generates a few tokens (the average length is 3.24 tokens in our experiments) for routing. The routing designs described in the Q4 of the rebuttal for reviewer gqRq also decrease the inference latency.

---

### Official Review · Reviewer_dPiw · 2024-07-14

**Soundness:** 3
**Presentation:** 3
**Contribution:** 2
**Rating:** 4
**Confidence:** 4

**Summary:**

This paper considers a critical issue in vision language models, i.e., the vision encoder design. The authors proposes MoVA, a coarse to fine routing strategy, adaptively routing and fusing task specific vision experts which best suit the task. Experiments conducted on a wide range of MLLM benchmark demonstrate the advantages of the proposed MoV adapter.

**Strengths:**

1.	The motivation is clear, and the method is easy to follow

2.	The proposed coarse to fine routing strategy is reasonable

**Weaknesses:**

1.	As for OCR and related tasks, it has been validated that high resolution is much more import than model size, and image split like LLaVA Next, Monkey, Ferret etc. have been widely used. I wonder the importance of including task specific models. It looks redundancy, and importantly, we cannot include all models that cover different tasks exhaustively

2.	As shown in Table 1, the main improvement results from the OCR related Doc and Chart task, which may resolved by current MLLMs baseline such as LLaVa Next, I wonder the  necessity of the using MoE experts, as other module, adding resolution is more reasonable. In Table 3, since the PT, SFT and pretrained models are different, it is hard to validate the gains are from the MoV experts.

3.	Does the method introduce error propagation, say, the first stage LLM routing is not correct, which may harm the performance of the whole framework. On the other hand, if the coarse routing is simple, could we say that this stage we do not need to carefully design, and a tiny LLM or even rule based according to the prompt is enough? Since current framework seems complex using two LLMs, the comparison is not fair, and I wonder if there is a better routing strategy.

**Questions:**

The main concern is the rationality of using MoV on vision side, while two step routing is interesting, it adds the complexity (using two LLMs), and it is not clear the advantages of MoV over the latest MLLM benchmark.

---

> ### Author Rebuttal · Authors · 2024-08-07
>
> Dear Reviewer dPiw,
>
> Thanks for giving so many constructive suggestions for our paper, I will clarify the settings.
>
> **Q1: The rationality of using task-specific vision models.**
>
> 1. We have conducted extensive experiments to benchmark different vision encoders in Table 1. We find a single CLIP encoder cannot generalize to various image domains, such as text recognition tasks.
> 2. Besides, a task-specific model can achieve optimal performance in its proficient task. It is reasonable to incorporate task-specific vision experts to improve performances on their relevant tasks. We also find the "plain fusion" of multiple vision experts can bring performance gains to the baseline with a single CLIP encoder in Table 1.
> 4. Many previous works [1-5] have demonstrated that the CLIP encoder fails to generalize to some domains and the visual capacity of MLLM can be enhanced by introducing additional vision experts.
>
> [1] Eyes Wide Shut? Exploring the Visual Shortcomings of Multimodal LLMs. CVPR 2024.
>
> [2] VCoder: Versatile Vision Encoders for Multimodal Large Language Models. CVPR 2024.
>
> [3] SPHINX-X: Scaling Data and Parameters for a Family of Multi-modal Large Language Models. ICML 2024.
>
> [4] SPHINX: The Joint Mixing of Weights, Tasks, and Visual Embeddings for Multi-modal Large Language Models. ECCV 2024.
>
> [5] Vary: Scaling up the Vision Vocabulary for Large Vision-Language Models. ECCV 2024.
>
>
> **Q2: We cannot include all models that cover different tasks exhaustively.**
>
> 1. We agree with this idea but we do not aim to exhaustively incorporate extensive vision encoders to cover all the tasks in this paper.
> 2. Instead, we aim to reveal the common limitations of previous mixture of vision experts methods and propose a new paradigm. This paradigm provides a new perspective for the community that we can dynamically incorporate relevant vision experts for specific tasks with poor performances in real-life scenarios.
>
>
>
> **Q3: Adding resolution is more reasonable.**
>
> 1. We agree that enlarging the image resolution can bring performance gains on OCR tasks. We also find **both adding resolution and adding vision experts (MoVA) are orthogonal approaches to enhance visual capacity in our experiments.** For example, we train a MoVA-7B model with the image split technique of LLaVA-NeXT (MoVA-HD) and compare it with the baseline on OCR tasks. Specifically, we divide an input image into 4 patches. The MoVA-HD can improve the score on the DocVQA benchmark from 81.3 to 84.0. This indicates such a high resolution setting also benefits MoVA.
> 2. The experiments in Table 1 have demonstrated the text recognition expert, such as Pix2Struct, can surpass the CLIP encoder on text-oriented VQA benchmarks by clear margins with aligned image token settings.
> 3. As described in Q2 of our global response, MoVA can achieve superior performance to LLaVA-NeXT with high resolution technique under the same settings.
>
>
> **Q4: It is hard to validate the gains are from the MoV experts.**
>
> Please refer to the Q2 of our global response.
>
> **Q5: Does the method introduce error propagation?**
>
> The incorrect routing results may harm the performance of the MLLM. In some cases, we find the multimodal gating in MoV-Adapter can assign low scores to these experts that are not relevant but incorrectly activated. We release some routing cases in the pdf file of global response.
>
> **Q6: Routing strategy design.**
>
> 1. In the Table 14 of paper appendix, we discuss the effect of LLM for expert routing. Specifically, we compare the LLM router with BERT router and MLP router. For smaller router variants, they require more routing training data and fail to attain comparable performance.
> 2. We only use 1 LLM for expert routing and response generation. Using a tiny LLM in the routing stage can bring addtional model parameters.
> 3. To ablate the routing performance of tiny LLM router (InternLM2-1.8B) and standard LLM router (Vicuna-7B), we mutually label a subset of our selected 500 routing evaluation samples as "easy" or "hard". The routing performance of MoVA on these 500 samples have been discussed in Appendix A.3. Finally, We obtain 50 easy samples and 20 hard samples. The routing accuracy is obtained by human evaluation. As shown in the table, the tiny LLM router achieves comparable accuracy on simple evaluation samples while failing to catch up to the standard LLM router on hard evaluation samples.
>
> | Router | Simple | Hard |
> | - | - | - |
> | InternLM2-1.8B | 96% | 55% |
> | Vicuna-7B | 98% | 80% |

---

> > ### Author Response · Authors · 2024-08-14
> > **Looking forward to your post-rebuttal feedback**
> >
> > Dear Reviewer dPiw:
> >
> > We thank you for the precious review time and valuable comments. We have provided corresponding responses and results, which we believe have covered your concerns. We hope to further discuss with you whether or not your concerns have been addressed. Please let us know if you still have any unclear parts of our work. If your concerns have been well addressed, please consider raising your rating, thanks.
> >
> > Best,
> >
> > Authors

---

### Official Review · Reviewer_Lntx · 2024-07-14

**Soundness:** 3
**Presentation:** 3
**Contribution:** 3
**Rating:** 6
**Confidence:** 3

**Summary:**

In this paper, the authors propose MoVA, a powerful MLLM composed of coarse-grained context-aware expert routing and fine-grained expert fusion with MoV-Adapter. Based on multimodal context and model expertise, MoVA fully leverages representation from multiple context-relevant vision encoder experts flexibly and effectively while avoiding biased information brought by irrelevant experts. MoVA can achieve significant performance gains over current state-of-the-art methods in a wide range of benchmarks.

**Strengths:**

1. The authors comprehensively investigate various vision encoders and analyze the performance of individual vision encoders versus the plain fusion of multiple encoders across various tasks.
2. The authors propose MoVA, a powerful MLLM composed of coarse-grained context-aware expert routing and fine-grained expert fusion with MoV-Adapter.
3. The proposed MoVA can achieve significant performance gains over state-of-the-art methods in a wide range of challenging benchmarks.

**Weaknesses:**

1. Despite the proposed MoVA in this paper achieving state-of-the-art performance across multiple tasks, leveraging multiple visual encoders in MLLM design does not introduce novelty. Additionally, the design of MoV-Adapter also lacks innovation.
2. This paper lacks a comprehensive evaluation regarding the routing selection of multiple visual encoders using LLMs, necessitating a more targeted and comprehensive assessment from the authors.

**Questions:**

Please refer to Weakness Section for more details.

**Limitations:**

The authors have clearly illustrated the weakness of this work by pointing out the hallucination and the model performance may be affected by failure cases of the context-relevant vision experts.

---

> ### Author Rebuttal · Authors · 2024-08-07
>
> Dear Reviewer Lntx,
>
> Thank you for appreciating our approach. We will address your concerns below.
>
> **Q1: Leveraging multiple visual encoders in MLLM design does not introduce novelty. Additionally, the design of MoV-Adapter also lacks innovation.**
>
> Leveraging multiple visual encoders is not our novelty. Please refer to the Q1 of our global response.
>
> **Q2: This paper lacks a comprehensive evaluation regarding the routing selection of multiple visual encoders.**
>
> 1. We have ablated the design of routing selection in Table 9, Table 10, and Table 14. We verify the effectiveness of our routing data in the line 598.
> 2. We present some routing cases in the pdf file of the global response.
> 3. We calculate the activation probability for each expert on several benchmarks:
>
> | Benchmark | DINOv2 | Co-DETR | SAM | Vary | Pix2Struct | Deplot | BiomedCLIP |
> | - | - | - | - | - | - | - | - |
> | MME | 63.6% | 12.9% | 2.8% | 26.2% | 29.3% | 28.5% | 0% |
> | MMBench | 54.6% | 15.8% | 3.2% | 21.9% | 22.6% | 22.6% | 0% |
> | DocVQA | 0% | 0% | 0% | 97.9% | 99.8% | 98.3% | 0% |
> | POPE | 100% | 100% | 0% | 0% | 0% | 0% | 0% |

---

> ### Comment · Reviewer_Lntx · 2024-08-14
> **Response to Authors**
>
> After checking the peer review comments and the author's responses, I decided to maintain the given score for this work.

---

> > ### Author Response · Authors · 2024-08-14
> > **Thanks for your comments**
> >
> > We deeply thank you for the kind support of our work!

---

### Author Rebuttal · Authors · 2024-08-07

Global response: We would like to extend our sincere gratitude to all reviewers for devoting their valuable time and expertise to reviewing our paper. We are delighted to hear that the reviewers have generally acknowledged and appreciated the contributions made in our work, which include

1. Clear motivation (dPiw) and reasonable method (dPiw, gqRq).
2. A new perspective to improve the reasoning ability of MLLMs (FVeU).
3. Comprehensive and extensive experimental results (Lntx, FVeU).
4. State-of-the-art performance (Lntx).


We express our sincere appreciation to all the reviewers for their insightful suggestions. To address the reviewers' concerns, we provide the following summaries and discuss them in rebuttal:

1. Paper novelty (Lntx, FVeU).
2. Performance comparison (dPiw, gqRq).
3. Inference cost (gqRq, FVeU).
4. Expert routing design and evaluation (Lntx, dPiw).
5. Necessity of vision experts (dPiw).

We will address the raised concerns and incorporate these improvements into our paper once we can edit it on OpenReview.
We make a global response for the Q1, Q2, Q3:

**Q1: The novelty of our paper**

1. Given $N$ vision encoders, previous works of the community all focus on how to directly fuse features generated by these $N$ **fixed mixture of vision experts** to solve every task without considering model expertise.
2. We provide a new perspective that we can first leverage the reasoning and tool-use ability of LLM to select $K$ ($K<N$) vision experts that are most relevant to the current task and then fuse these $K$ **dynamic mixture of vision experts**.
3. We conduct comprehensive experiments to reveal that the feature fusion of fixed mixture of vision experts fails to achieve the optimal performance and leads to performance degradation if we do not consider the relations between model expertise and the current task.

**Q2: Performance comparison using the same data**

We conduct experiments on LLaVA-1.5-7B and LLaVA-NeXT-7B with the same data and LLM as MoVA-7B. The performance comparison can be found in the pdf file of the global response.
As presented in the table, our method still outperforms LLaVA-1.5 and recent state-of-the-art LLaVA-NeXT on a wide range of benchmarks under the same settings.
More importantly, MoVA with 576 image tokens can beat LLaVA-NeXT with image split technique and high-resolution input image on text-oriented benchmarks.


**Q3: Inference analysis**

As illustrated in Figure 1 of our paper, MoVA consists of two stages: coarse-grained context-ware expert routing and fine-grained expert fusion with MoV-Adapter. This two-stage inference pipeline can be further broken down into 5 steps:
1. (Stage 1) Data preprocessing. We first process the input image with image processors and convert the input text into a token sequence with the LLM tokenizer.
2. (Stage 1) Base encoder forward. We extract the base image feature using the base CLIP encoder. Note that we only run the base encoder once since its output feature can be preserved and reused in the fourth step.
3. (Stage 1 ) LLM routing generation. We compress the base image features into 144 image tokens. The LLM generates a concise routing answer based on the compressed image feature and routing instruction.
4. (Stage 2) Vision experts and MoV-Adapter forward. According to the multimodal context and routing results generated in the previous step, we fuse vision features of the base encoder and activated experts in a coarse-to-fine manner.
5. (Stage 2) LLM response generation. The LLM generates the final response given the fused vision features and user instructions.

To investigate the inference efficiency of each step, we randomly select 200 images from the COCO val2017 dataset and adopt the common image caption instruction: *Describe this image*. The temperature for generation is 0. The latency is measured using bfloat16 and flash-attention 2 on an A100 80G GPU. The benchmark results of model inference are presented here:

| Step 1 | Step 2 | Step 3 | Step 4 | Step 5 | Routing output | Final response |
| - | - | - | - | - | - | - |
| 0.19s | 0.05s | 0.14s | 0.07s | 10.24s | 3.24 tokens | 405.06 tokens |

We present the average inference latency of each step and show the average sequence length of the routing output and final response.
**Compared to the LLM response generation (Step 5), the LLM expert routing (Step 3) generates much fewer output tokens and its latency is negligible (0.14s *vs* 10.24s).** Therefore, our method does not bring significant inference costs.

Moreover, we compare the inference efficiency of MoVA with the common **fixed mixture of vision experts** paradigm. We use DINOv2, Co-DETR, SAM, Vary, Pix2Struct, Deplot, and BiomedCLIP as the mixture of vision experts. Specifically, we do not present the latency of LLM final response generation (Step 5) since the only difference is the visual component.

| Mixture of vision expert | Step 1 | Step 2 | Step 3 | Step 4 | Total |
| - | - | - | - | - | - |
| MoVA | 0.19s | 0.05s | 0.14s | 0.07s | 0.45s
| Fixed | 0.19s | 0.05s | N/A | 0.43s | 0.67s |

The results reveal that dynamically selecting the most $K$ relevant experts from $N$ experts can accelerate model inference while boosting performances.

We also compare the memory usage of MoVA and MoVA without expert routing. As shown in the following table, the LLM routing only increases the memory usage from 25165 MB to 27667 MB.

| MoVA | MoVA w/o routing |
| - | - |
| 27667 MB | 25165 MB |

---

### Decision · Program_Chairs · 2024-09-25

**Decision:**

Accept (poster)

**Comment:**

This paper introduces a new MLLM called MoVA, which is composed of coarse-grained context-aware expert routing and fine-grained expert fusion with MoV-Adapter. By utilizing multimodal context and model expertise, MoVA fully leverages representation from multiple experts. The experimental results validate the model's effectiveness across multiple benchmarks.

The paper was evaluated by four reviewers. Three reviewers recommended acceptance (1x accept, 1x borderline accept, 1x weak accept), while one reviewer suggested a borderline rejection. Overall, this work presents a novel and effective MLLM solution that has the potential to inspire further research in the field.

The authors are encouraged to incorporate their responses to the reviewers' comments in the revised version to enhance the paper's clarity and comprehensiveness.